# Setting up the Largest Mass Vaccination Center in Europe: The One-Physician One-Nurse Protocol

**DOI:** 10.3390/vaccines11030643

**Published:** 2023-03-13

**Authors:** Vanessa Houze-Cerfon, Benoit Viault, Léa Zerdoud, Marie Ged, Sébastien Vergé, Florence Metz, Gregory Ciottone, Alexander Hart, Vincent Bounes

**Affiliations:** 1SAMU 31 (Service d’Aide Médicale d’Urgence Haute-Garonne), Disaster Medicine Unit, Toulouse University Hospital, 31059 Toulouse, France; 2Agence Régionale de Santé (ARS) Occitanie, 31000 Toulouse, France; 3Service Départemental d’Incendie de de Secours (SDIS) Haute Garonne, 31100 Toulouse, France; 4Department of Emergency Medicine, Beth Israel Deaconess Medical Center, Boston, MA 02215, USA; 5Department of Emergency Medicine, Hartford Hospital, Hartford, CT 06106, USA; 6School of Medicine, University of Connecticut, Farmington, CT 06032, USA

**Keywords:** mass vaccination, COVID-19, disaster medicine, health planning organizations

## Abstract

To manage mass vaccination without impacting medical resources dedicated to care, we proposed a new model of Mass Vaccination Centers (MVC) functioning with minimum attending staffing requirements. The MVC was under the supervision of one medical coordinator, one nurse coordinator, and one operational coordinator. Students provided much of the other clinical support. Healthcare students were involved in medical and pharmaceutical tasks, while non-health students performed administrative and logistical tasks. We conducted a descriptive cross-sectional study to describe data concerning the vaccinated population within the MVC and the number and type of vaccines used. A patient satisfaction questionnaire was collected to determine patient perception of the vaccination experience. From 28 March to 20 October 2021, 501,714 vaccines were administered at the MVC. A mean rate of 2951 ± 1804 doses were injected per day with a staff of 180 ± 95 persons working every day. At peak, 10,095 injections were given in one day. The average time spent in the MVC was 43.2 ± 15 min (time measured between entry and exit of the structure). The average time to be vaccinated was 26 ± 13 min. In total, 4712 patients (1%) responded to the satisfaction survey. The overall satisfaction with the organization of the vaccination was 10 (9–10) out of 10. By using one attending physician and one nurse to supervise a staff of trained students, the MVC of Toulouse optimized staffing to be among the most efficient vaccination centers in Europe.

## 1. Introduction

Successive waves of coronavirus-2019 disease (COVID-19) have impacted countries worldwide since its emergence in December 2019. Since efficient and safe vaccines were developed, raising population immunity levels through rapid vaccination coverage became paramount to decrease morbidity and mortality [1].

The vaccination campaign started in France on 27 December 2020 once the Cominarty^®^ and Vaxzevria^®^ vaccines received marketing authorization from the European Medicine Agency [2]. The first phase of the vaccination program used multiple closed points of dispensing (hospitals, small vaccination clinics, family medicine clinics, etc.) through a centralized appointment management platform. This ensured the administration of vaccines to vulnerable populations while vaccine supplies were still limited, and production was uncertain. 

Mass vaccination centers (MVC), defined as “points of massive dispensing of vaccines to a large share of the general population in response to an outbreak of a contagious disease”, are acknowledged as the best solution to administer vaccines in the shortest time to the greatest number of people [3,4]. An early MVC in a decontamination unit was deployed in Toulouse on 6 and 7 March 2021 and vaccinated 1700 people [5]. As vaccine supplies became more stable, the Center for Disaster Response (CRC) of Toulouse decided to implement a mass vaccination center at the Toulouse Metropole exhibition hall beginning on 28 March 2021. Although numerous mass vaccination campaigns occurred in France in the past few decades (H1N1 influenza in 2009 or seasonal influenza), few have had their organization and impact documented in the literature. Unlike other existing MVC, Toulouse’s “vaccinodrome” (as it became dubbed) had the peculiarity of operating with only one physician, one nurse, and a team of trained students, thus freeing up healthcare personnel for crisis management in healthcare facilities. Principles of disaster medicine were used in the design of the center (forward flow, task simplification, and specific training in high-risk conditions) to ensure appropriate care despite fewer resources than normal [6]. This article details the results (and experience drawn) from the functioning of this MVC through an observational study in order to propose a new model of MVC functioning with a minimum attending staff.

## 2. Materials and Methods

### 2.1. Design

We performed a descriptive, cross-sectional study conducted as an intensive, systematic investigation of a new model in which we examined in-depth data relating to several variables. It took place in the Toulouse MVC, from 28 March to 20 October 2021. The main outcomes of this study were the number of doses administered, the number of staff and caregivers involved, and most importantly the ratio between doses and vaccinators. As secondary outcomes, we collected characteristics of vaccinated people, the length of stay, and the vaccination process duration. The number of immediate major side effects were analyzed, defined as death, life threatening or hospitalization. Mid and/or side effects were not recorded. Through the quality and satisfaction survey, people satisfaction regarding different aspects of care were recorded. 

### 2.2. Organization of the Vaccination Process in Toulouse

#### 2.2.1. Premises 

The Toulouse urban area is a 5381 km^2^ area located in the south of France, with around 1.4 million inhabitants. 

The vaccination operation took place in the exhibition center of Toulouse (made available by the city) given the need for substantial capacity, a central location, and accessibility. The overall surface area of the building was 7000 m^2^ in two separate buildings, it had an entrance and a separate point of egress so as to allow for continuous forward motion flow, as reversing the flow of crowds has the potential to cause disruptions in disaster medicine situations [7]. The MVC was open 7 days a week from 8 a.m. to 7 p.m., from 28 March to 20 October 2021. During 2 15 day-periods in summer, the opening hours were extended to 10 p.m.

#### 2.2.2. Staff

MVC operations were based on an inter-service collaboration under the aegis of the Disaster Response Center (CRC), a grouping of healthcare institutions, local authorities, the firefighter service, and pre-hospital care linked to the disaster response.

The assignment of roles and responsibilities for each service involved in the MVC was predefined in an organizational chart similar to a classic incident command system (Figure 1).

The MVC was under the supervision of one physician medical coordinator, one nurse coordinator, and one operational coordinator per 6-h shift:

The **medical coordinator** was an emergency physician or an anesthesiologist trained in disaster medicine. He/she was involved in the medical triage and management of post-vaccination adverse events during the surveillance period. He/she ensured the proper functioning of the vaccination process, checked the number of vaccination lanes, and participated in the briefing and debriefing of the teams;The role of **the nurse coordinator** consisted of ensuring the logistical management of the MVC (vaccines, equipment supplies, etc.) and in managing the vaccinators and preparers. He/she was in charge of keeping track of the preparation, quality, distribution, and flow of the doses using a follow-up sheet. He/she was also responsible for monitoring the refrigerators in the preparation area and checking the number of remaining doses at the end of the day;The **operational coordinator** oversaw the briefing and debriefing of the staff at the start and the end of each shift. This role was usually assigned to a firefighter officer due to their knowledge of command and management;Students were the most important part of human resources. **Healthcare students** were involved in triage (medical and pharmacy students), pharmaceutical control and vaccine preparation (pharmacy students), vaccine injection (medical and nurse students), post-vaccination surveillance (all healthcare students), and adverse effect observation (medical and nurse students) while **non-healthcare students** were involved in administrative work and logistics, still under the supervision of particularly experienced students that were designated as station managers (in charge of one step of vaccination) or team leaders (in charge of all station managers in the MVC). 

Eventually, some volunteers from the local first aid association and firefighters under the supervision of the operational coordinator were involved in the different steps of the chain according to their professional skills.

The required personnel/shot ratios were defined empirically in the first months of operation, adapted in real-time and are mentioned for reference in Table 1. 

#### 2.2.3. Spatial Organization

To set up the MVC, the coordinating team followed the “forward motion flow” doctrine used in the national smallpox threat response plan [8]. It is an organization designed to create successive stations forming a vaccination line with no reverse motion (Figure 2).

At the entrance, people presenting for vaccination had to show their appointment confirmation and received an informed consent letter on vaccination, including a questionnaire detailing any contraindications. Once signed, they were directed to temperature control, where medical and pharmaceutical students checked the contraindications questionnaire. The next step was the verification of the immunization schedule with the help of the national registration software. If a contraindication was found, the vaccination was canceled and another appointment was immediately made when applicable, or an orientation to a specialist was scheduled if needed. 

Patients without contraindications were directed to the entrance administrative checkpoint, where their appointment was validated in a web-based appointment management solution (Keldoc, NEHS DIGITAL SAS, Malakoff, France) and then orientated to an available lane to be vaccinated. The number of lanes was adapted according to the vaccination needs of the population and could reach a peak of 64 lanes at maximum activity. The vaccination was then recorded in a national institutional database for vaccination management (AMELI, caisse nationale de l’assurance maladie, Paris, France) at the exit administrative checkpoint, and follow-up appointments were taken in Keldoc for booster shots if needed. Early side-effect onsets were monitored in the post-vaccination surveillance zone: 15 mn for most patients and 30 mn when risk factors were identified (uncontrolled hypertension, ongoing pregnancy, and anticoagulant drug use).

All these zones were physically demarcated via crowd management equipment (barriers and caution tape) and markings (floor lines, noticeboards, and wall signs), and were oriented in a fashion so as to encourage forward movement and not permit a way back. 

All stations and coordinators were equipped with radios set to a local network to aid in communication.

A specific line was dedicated to the vaccination of disabled people, accompanied by dedicated staff members.

#### 2.2.4. Logistics

The management of logistics and the supply chain was entrusted to non-health students working in close collaboration with suppliers and the hospital pharmacy to organize the storage of supplies and ensure that all zones were provisioned in real-time. During the opening period of the MVC, the two vaccines BNT162b2 (Cominarty®, Pfizer-BioNTech’s, New York, NY, USA) and mRNA-1273 (Spikevax®, Moderna™, Cambridge, MA, USA), were used according to the supply made available by the health authorities. The lanes and preparation areas were well identified and separated according to the vaccine. Doses were reconstituted by pharmacy students in separate locations. A triple control organization of the prepared doses was set up. A student prepared a vial and the quality of the syringes was controlled by another student. The ready-to-use syringes were then carried to the injectors in trays of 7 or 10 doses. The third control was performed by the injection administrator when he/she was delivering the vaccine. 

#### 2.2.5. Training

Students were assigned to each position in the vaccination process after receiving appropriate training for that task. This training consisted of 3-h theoretical courses intended to remind the students about the virus and to present the general functioning of the system as well as the roles of each person. Separately, each student was trained for their position according to their field of study and skills during a half day of practical training. Station managers and team leaders were chosen amongst the more experienced students and went through an accreditation of their experience organized by the training/quality team before assuming their role and were accredited by the medical coordinator to administer vaccines.

### 2.3. Data Collection

Data concerning the population vaccinated and the number and type of vaccines used were gathered from Keldoc and AMELI databases and the MVC records.

Data concerning the MVC personnel were collected from the MVC web-based planning tool (Planning-Medical.com, Planning-Services.fr SARL).

A written satisfaction questionnaire was available in the center for any patient who wished to give feedback about their experience. The questionnaire was freely available in the center, but not required from patients. An information letter, with the subject, objectives, and methodology of the study was provided with the questionnaire and the patients were asked to read it and give oral consent prior to participating in the study. Informed consent was waived because ethical approval was not applicable according to French ethic and regulatory law, namely article R1121-1 of the public health code. 

The survey was divided into four parts, including (1) sociodemographic characteristics (age, gender, and job status) and distance from a household, (2) the existence of risk factors and type of vaccine selected, (3) the total time needed for the vaccination process (difference between entrance in the MVC and exit time), the targeted time until vaccination and satisfaction of the participants regarding time requirements, and (4) the self-report assessment by closed-ended questions on the confidentiality, accessibility, safety, quality of reception, and information in the MVC. The questionnaire used a global descriptive scale, scored from 1 (complete dissatisfaction) to 6 (complete satisfaction). At the end of the survey, a global satisfaction feedback evaluation was also asked, based on a score rated from 0 (completely dissatisfied) to 10 (completely satisfied). People willing to complete the survey were interviewed by students, who entered the data of the questionnaire online. It was explained to them that this was anonymous and could be used for research purposes. The data collected by the questionnaire were not linked to the vaccination data.

### 2.4. Statistical Analysis

All analyses were conducted using Stata version 14 (StataCorp, LLC, College Station, TX, USA). The collected data were double-checked by the research assistant and the data manager of the research staff. Descriptive statistics were applied to calculate the proportions, frequencies, and confidence intervals as well as the means and standard deviations or the medians and interquartile ranges for ordinal variables.

## 3. Results

### 3.1. Data on the MVC Process

From 28 March to 20 October 2021, 501,714 vaccines were administered at the MVC. A mean rate of 2951 ± 1804 doses was injected per day with a staff of 180 ± 95 persons working every day to ensure a constant vaccination flow. There was a mean number of 225.3 ± 62.3 vaccines administered per vaccinator and per day. The peak was reached on 24 July, with 10,095 injections in one day (Figure 3). 

It is notable that 503,667 doses were prepared, meaning only 1953 doses (0.39%) were thrown away. In total, 3077 people worked in the MVC during 6 or 10 h shifts. Almost all of those were students, participating according to their training and university schedules. The average time spent in the MVC was 43.2 ± 15 min (time measured between entry and exit of the structure). The average time to be vaccinated was 26 ± 13 min (minutes from the entry point to being vaccinated).

During the entire MVC period, only five ambulances were called to respond: two for symptoms detected prior to vaccination (bradycardia and chest pain) and three for post-vaccination symptoms (one each of chest pain, hypertension, and vasovagal syncope). All those patients were evacuated for evaluation in the emergency department. The cumulative incidence of side effects over the study period was 0.996 per 100,000.

### 3.2. Data on the Subjects That Were Questioned about Their Satisfaction with the Process

The quality and satisfaction survey was filled out by 4712 people, with a response rate of 0.94%. The survey respondents had a mean age of 47 ± 15 years and were more represented by women (54.9%) than men (45%). Most participants were immunized with the Cominarty^®^ vaccine (88.1%, *n* = 4143) followed by the Spikevax^®^ vaccine (11.8%, *n* = 558) and most had no risk factors (87.6%, *n* = 4088) (Table 2).

### 3.3. Data on Satisfaction

Participants were satisfied with the duration of time in the facility (88.5%, *n* = 4163). Participants felt perfectly safe about the hygiene rules in 87.5% (*n* = 3983) of cases and 84.7% felt their confidentiality was completely respected (*n* = 3852). Most of the participants expressed complete satisfaction regarding staff availability (96.1%, *n* = 4371) and the quality of information given (78.7%, *n* = 3564). The MVC was felt to be perfectly accessible for 78.9% (*n* = 3704) of participants (Table 3). The overall satisfaction with the organization of the vaccination was 10 (9–10) out of 10.

## 4. Discussion

Toulouse set up a huge mass vaccination center during the vaccination period that has proven to be, to our knowledge, the biggest center in Europe with a total of 501,714 doses of vaccine delivered over 7 months, which represents 0.5% of the total vaccines delivered in France as of 20 October 2021 (a total of 96,061,990 doses administered) [9]. France hosted 7291 vaccination centers, with an average number of injections of 13,175 per center. Some notable examples from the literature include the conversion of EuroDisney Paris in the spring of 2021 to an MVC with a potential capability of 1000 doses/day, while the Health Ministry of Quebec in its guidelines specifically targeted 2500 doses/day, and during May 2021, MVCs from Italian high and medium-sized cities were able to immunize over 4000 subjects in a single day [10,11,12]. Moreover, thousands of family physicians and nurses throughout France performed vaccinations outside of these mass vaccination centers.

One of the biggest values of the MCV in Toulouse has to do with the inclusion of healthcare and non-healthcare students in the vaccination effort. This paper is an opportunity to describe what enabled this solution in Toulouse and to show that despite using this additional resource, avoiding depleting other care services is an option in epidemic emergencies. Disaster medicine differs from other disciplines since it requires unique working conditions and principles not found in normal health practices. Amongst the principles, the ability to implement, command, control, and coordinate outside of the medical discipline is widely discussed [13]. Disaster medicine incorporates many characteristics of management science such as command, coordination, planning, and strategy, but dealing mainly with students implies strong training, standardizing job descriptions, and closely performance monitoring. During the mass vaccination that took place more or less all over the Global North, the lack and/or diversion of human resources from acute care to vaccination centers was one of the major challenges and in most situations, the use of other resources such as healthcare students was not accepted (in some cases even the use of healthcare workers that usually do not vaccinate was a challenge) [14].

Given the limited resources and healthcare personnel in many regions of the world, some authors have recommended the use of lean principles and continual learning. These approaches can be learned conveniently through large open online courses and can be used to create high throughput vaccination sites, as described by Froman et al. [15]. With 78 personnel working efficiently and effectively together, they described a maximum throughput of 5024 injections over 10 h. As compared with other published COVID-19 mass-vaccination sites, this study described a threefold to fourfold higher productivity than most other published settings. In another study by Phillips et al., 640 vaccinations per hour were performed using 70 people at a time, including 36 nurses [16]. The Israel Defense Forces reported recruiting military medics and paramedics who were trained to administer the vaccines [17].

Other authors report massive recruitment of nurses, medics, and paramedics for the administration of COVID-19 vaccines [18,19]. Thus, the ability to increase the recruitment of dedicated healthcare professionals is an important factor in vaccination success. Some countries have had to divert health professionals from acute care, a highly undesirable necessity at a time when many health systems were being overwhelmed with patients suffering from the acute effects of COVID-19 [3]. A recent literature review by Gianfredi et al. described the type and/or competencies of staff needed in a mass vaccination center. In particular, the most frequently reported were physicians, nurses, and pharmacists among medical staff, whereas traffic flow personnel, data collection personnel, and volunteers were among the most frequently required nonmedical staff. Our center, with its organization, has the advantage of using very few attending caregivers, as only one physician and one nurse were needed at any given time. Students were prepared to perform the vaccinations tasks via a mandatory training session. The staff were then divided into teams, with each headed by a more experienced student who supervised the others.

The healthcare system must be able to increase its capacity through continuous flexible adaptation, or its ability to manage an increased number of patients [20]. Using students to provide vaccines allows the efficient rollout of a vaccination campaign while diverting a minimal number of trained healthcare providers away from critical acute care tasks.

One obvious factor in the success of COVID-19 vaccination campaigns is the presence of mass vaccination plans. The implementation of a mass vaccination program is a difficult task, requiring coordination between multisectoral teams of professionals, complex logistics, and the limited availability of vaccine doses and caregivers. As a consequence, every participating organization had to face heavy challenges to reach the goals of the vaccination programs in the shortest time and the safest way possible [3,14]. 

There is much to be learned from the experiences of those at the forefront of the vaccination program rollouts, even if the literature concerning this topic is still relatively sparse [18,21,22]. Despite slight differences, the most common aspects of each of these centers were the entrance, registration, waiting rooms (in many cases with an educational video), screening/anamnesis, vaccination room, post-vaccination observation room, and exit. 

Our MVC implemented a multi-organizational response in the unstable work environment based on the expertise of prehospital teams, with the help of various medical first response equipment, as our teams are specialized in field hospital establishment and are specially trained to save lives in high-risk conditions. The MVC organization allowed a forward motion flow, where the patients entered the vaccine area, get a formal identification, aretriaged (e.g., confirmation that the patients belonged to the prioritized groups), get the vaccine shot, and were directed to a waiting and discharge room [5].

As we implemented a center with two graduated caregivers, our productivity is off limits. Of course, our MVC took place in a large university city (where significant student manpower is available), so it may not be reproducible in smaller cities or those without a local university. As such, appropriate legislation is necessary for students to be authorized to vaccinate [23], which was passed by the French government during the early stages of the vaccination campaign.

Moreover, in a period of economic crisis, and with a shortage of part-time jobs for students, this MVC model financially supports the local student community while having them support an exhausted healthcare system.

### Limitation

We obtained little data on patients themselves, as it was felt that patient characteristics would be less pertinent to those attempting to replicate this MVC model. Lastly, the satisfaction and quality survey were only completed by roughly 1% of all patients, and a mandatory satisfaction survey may capture some patient opinions which were missed in this study.

## 5. Conclusions

By training and utilizing students to provide vaccinations, the MVC of Toulouse has demonstrated a new model of MVC. This model optimizes human resources during a public health crisis by requiring only one physician and one nurse to be present on the scene, thus preventing the diversion of scarce resources away from acute care centers.

## Figures and Tables

**Figure 1 vaccines-11-00643-f001:**
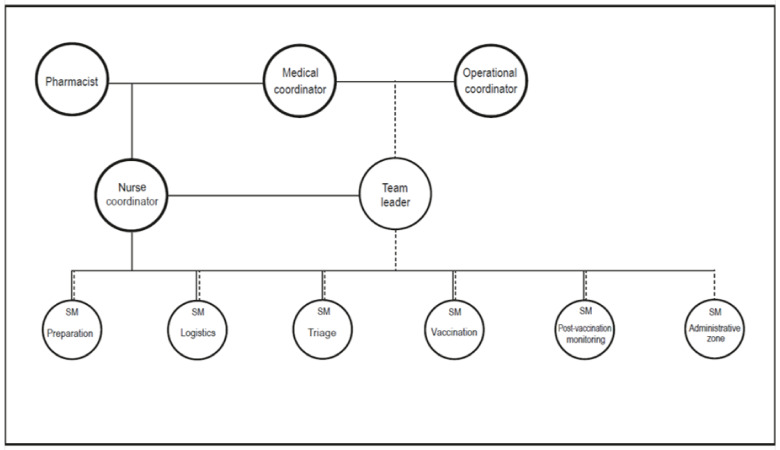
Toulouse Mass Vaccination Center command system. Full line – medical and para-medical supervision; dot line—administrative supervision.

**Figure 2 vaccines-11-00643-f002:**
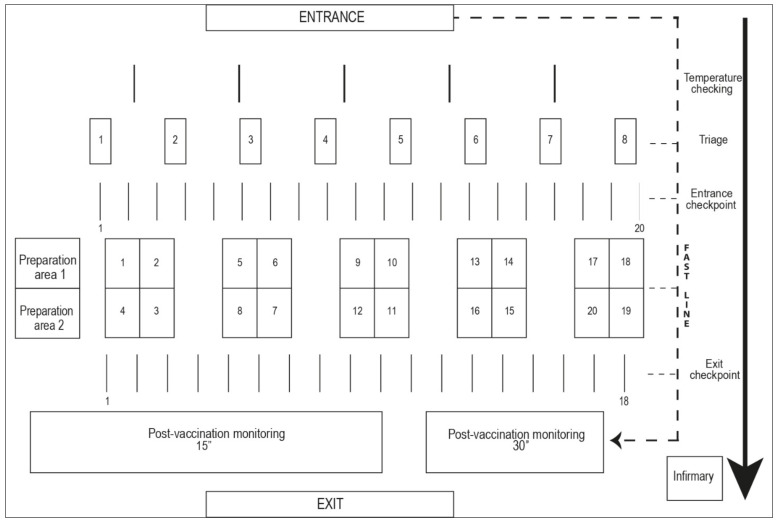
Toulouse Mass Vaccination Center geographical design. Every vertical line represents a station at each step of the process. For example, there are five stations at the temperature checking. Numbers correspond to the different stations in a specific area.

**Figure 3 vaccines-11-00643-f003:**
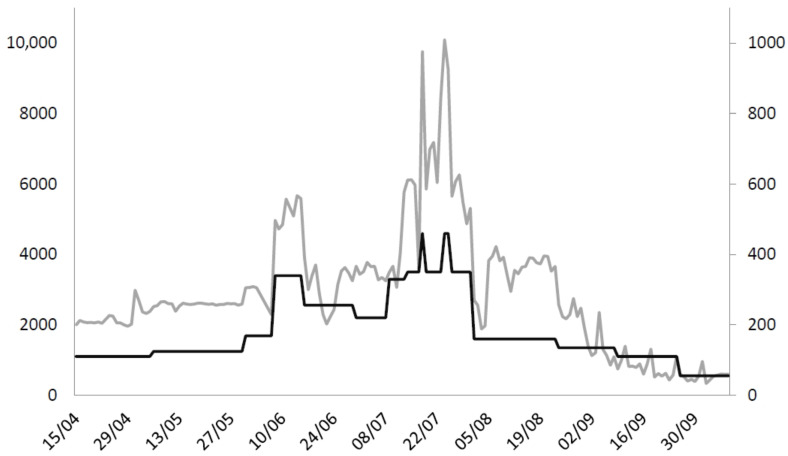
Number of daily doses and staff in the Toulouse Mass Vaccination Center. Vertical left axis and grey line: number of daily doses; vertical right axis and black line: number of staff members.

**Table 1 vaccines-11-00643-t001:** Number of personnel required per shot administered in a day (10 h), in each zone.

Zone	Personnel/Shot
Contraindication and temperature checking	1/450
Medical triage	1/250
Administrative zone 1 (entrance checkpoint)	1/150
Preparation area	1/300
Vaccination boxes	1/180
Administrative zone 2 (exit checkpoint)	1/200
Post-vaccination surveillance	1/500

Mean values for the entire time period.

**Table 2 vaccines-11-00643-t002:** Demographic characteristics of survey respondents.

Characteristics	Vaccinated Population *n* = 4712
Gender, *n* (%)	
Male	2116 (45)
Female	2580 (54.9)
Mean Age (95%CI)	47 (46.5–47.4)
Vaccine administered	
Spikevax^®^	558 (11.9)
Cominarty^®^	4143 (88.1)
Risks factors	579 (12.4)
Socio-professional class	
Working	3324 (71.1)
Retired	751 (16)
Student	381 (8.1)
Unemployed	227 (4.8)
Distance from home to vaccination center (km)	
<5	1140 (22.1)
5–10	1178 (25)
10–20	1527 (32.5)
20–30	604 (12.8)
>30	353 (7.5)

**Table 3 vaccines-11-00643-t003:** Evaluation of the vaccination operation by the vaccinated population.

Questions	Strongly Satisfied *n* (%)	Somewhat Satisfied *n* (%)	Slightly Satisfied *n* (%)	Slightly Unsatisfied *n* (%)	Somewhat Unsatisfied *n* (%)	Strongly Unsatisfied *n* (%)
Duration of time in the facility	4163 (88.5)	446 (9.5)	28 (0.6)	13 (0.3)	10 (0.2)	42 (0.9)
Respect of hygiene rules	3983 (87.5)	537 (11.8)	16 (0.3)	9 (0.2)	4 (0.1)	1 (0.0)
Respect for confidentiality	3852 (84.7)	641 (14.1)	32 (0.7)	9 (0.2)	8 (0.2)	8 (0.2)
Staff availability	4371 (96.1)	161 (3.5)	14 (0.3)	1 (0.0)	2 (0.0)	1 (0.0)
Quality of information	3564 (78.4)	789 (17.3)	129 (2.8)	28 (0.6)	24 (0.5)	14 (0.3)
Accessibility	3704 (78.8)	850 (18.1)	76 (1.7)	34 (0.7)	25 (0.5)	8 (0.2)

Results are expressed as frequency and percentage.

## Data Availability

Data are available upon request from the authors.

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
