# Peer review of "Setting up the Largest Mass Vaccination Center in Europe: The One-Physician One-Nurse Protocol"

_vaccines, 2023, doi:10.3390/vaccines11030643_

Round 1

Reviewer 1 Report

This is a valuable study documenting the processes of a MVC and demonstrating a method of preserving health professionals in patient management roles etc, through using students. Documenting such activities will be crucial support for future such events, and potentially other different hazard events affecting large numbers. 

The only issue is that I do not see clear mention of ethics approval for the research as you were interviewing human participants. 

1. The research addresses the protocol of setting up a large Mass Vaccination clinic in Europe with minimal staffing and assesses the patient experience.

2. The research addresses a gap in the field in reporting the set up protocol of a mass vaccination clinic with minimal staffing. There is a need for more publications and documentation of such processes to ensure the knowledge is available for future pandemics. This was a real issue for this pandemic where despite the use of some mass vaccination clinics in previous pandemics the documentation of processes was limited. Different protocols were used in different countries and contextualised to the local populations and resources. I am grateful to see this protocol published.

3. This mass vaccination clinic is particularly notable for the use of students and the low medical staffing. This is valuable at a time when medical staff are managing with large numbers unwell patients.

4. An observational study was an appropriate design for answering the research question. The collection of quantitative and qualitative data and the use of a questionnaire as a tool to gather the data was appropriate. The methods are clearly described. The statistical analysis is clearly described and checked by two researchers. They correctly mention the limited percent of participants completing the questionnaire of 1% so drawing conclusions from such a low percentage affects the quality of the data and the strength of conclusions drawn.

5. The research addresses the main question and are consistent with the evidence.  I might question the statement line 239 statin: “One obvious factor in the success of Covid-19 vaccination campaigns has been the presence of mass vaccination plans” – this comment appears unsupported by the data as there is no discussion or data provided on other methods of vaccination in comparison. I know that in some countries General Practitioners delivered large numbers so I would suggest that to make their statement they need to have assessed the proportion of vaccinations delivered by other health care services c/w the mass vaccination clinics.

I think this article should be published as is with the additional of the information on the ethics and addressing my comments.

Author Response

Reviewer 1

This is a valuable study documenting the processes of a MVC and demonstrating a method of preserving health professionals in patient management roles etc, through using students. Documenting such activities will be crucial support for future such events, and potentially other different hazard events affecting large numbers. 

The only issue is that I do not see clear mention of ethics approval for the research as you were interviewing human participants.

Response: Thank you for your reviewing. We added a paragraph in the 2.3 section “An information letter, with the subject, objectives and methodology of the study was provided with the questionnaire and the patients were asked to read it and to give an oral consent prior to participating in the study. Informed consent was waived because ethical approval was not applicable according to French ethic and regulatory law, article R1121-1 of the public health code.”

The research addresses the main question and are consistent with the evidence.  I might question the statement line 239 statin: “One obvious factor in the success of Covid-19 vaccination campaigns has been the presence of mass vaccination plans” – this comment appears unsupported by the data as there is no discussion or data provided on other methods of vaccination in comparison. I know that in some countries General Practitioners delivered large numbers so I would suggest that to make their statement they need to have assessed the proportion of vaccinations delivered by other health care services c/w the mass vaccination clinics.

Response: We agree with your comments. We enriched the discussion section with data from other articles that support our statement.

Reviewer 2 Report

Dear Authors

Thank you for submitting your paper to the journal.

This paper reports the experience of setting up a MVC in Europe during the SARS-CoV-2 pandemic which I consider to be very relevant and important for future planning of this and similar interventions.

Nevertheless, the format chosen (research article) is not the most appropriate to clearly report the study. I would advise an adaptation to the case report format and I have made this suggestion to the editors. Similarly, I would advise you to consider this study as a case study instead of an observational study. A good reference for this type of study is Robert K. Yin. (2014). Case study research design and methods (5th ed.). Thousand Oaks, CA: Sage. 282 pages.

Nevertheless, the paper is well written, and it reads well. I consider that it should be considered for publication after major revision.

Congratulations and warm regards

Suggested revisions:

1.       In the abstract, as well as throughout the text the author use commas to separate thousands and for decimal places. It is common use in English language to use commas to separate orders of magnitude and dots to separate decimal places. Please review accordingly.

2.       Line 56-57 . From my point of view one of the main issue that should be addressed by this type of manuscript is to specify exactly how the services were organized and designed. I would advise to specify what is meant by "Principles of disaster medicine were used in the designing of the center to ensure appropriate care despite fewer resources than normal" - What principles? Give at least one example

3.       Line 63-64 "We performed a prospective observational study (both quantitative and qualitative) of the Toulouse MVC, from March 28th to October 20th, 2021." The authors should separate the type of study from the context where it was conducted. I advise to state to classify as an observation cross-sectional study since the outcomes were measured only once (althought pertaining to a period of time). Also clarify the qualitative part of the study.

4.       Move text in lines 66 to 69 "Data concerning the population vaccinated, the number and the type of vaccines used were recorded as part of the normal quality assurance processes of the MVC. A written satisfaction questionnaire was available in the centre for any patient who wished to give feedback about their experience." to 2.3. Data gathering and data management.

5.       The use of a), b) , c), … under “2.2. Organization…” is a little bit confusing Please check if you can use sub-headings like “2.2.1 Premises…” or avoid identations

6.       Line 82 – No need for Health to have a capital H

7.       Please note that Figure 1 is black and white. Consider changing the type of lines used instead of using colours (e.g., dot line, doubled lines).

8.       Line 97 – review “in managing the human resources of the vaccinators and preparers” to “in managing the vaccinators and preparers

9.       Line 101 – Due a paragraph break in line 101, before “The operational coordinator oversaw the…”

10.   Underline or use bold to identify, the first time they are used, medical coordinator, nurse coordinator, operational coordinator, health care students and non health care students (line 89 to112).

11.   Consider presenting the roles/functions of medical coordinator, nurse coordinator, operational coordinator, health care students and non-health care students using dots for easiness of read or considered presenting a table systematizing this information

12.   Line 122 to 123. The sentence “The personnel/shot ratio was defined empirically in the first months of operation and adapted in real time, and are mentioned for reference.” Breaks the reading flow. Please consider deleting or moving to a more appropriate place in the text

13.   Clarify where this was done Line 124-125 “Firstly, people presenting for vaccination had to show their appointment confirmation and received an informed consent letter on vaccination, including a questionnaire detailing any contraindications.” (Entrance?)

14.   Line 126-127 where it reads “Once signed, they were directed to temperature control, then medical and pharmaceutical students checked their questionnaires.” It should read “Once signed, they were directed to temperature control,
where medical and pharmaceutical students checked the contraindications questionnaire.

15.   Line 164 consider replacing “to inject vaccines” by “to administered vaccines”.

16.   Data gathering and data management – there are two different aspects in this paper – one that relates to the registers of the vaccination process and that the authors used later on in the text and the specific data collection of the study that pertains to a questionnaire on the quality/ satisfaction of users, not mandatory, and that the authors also use in the paper. It is important to distinguish between the two – reporting how the data was gather for the vaccination registers and how the data from the MVC was used for the purpose of this study.

17.   The authors should state if an informed consent for the satisfaction questionnaire was obtained as well as if the study was assessed by an ethics committee. It is also important to stress that the questionnaire was anonymous (which the author already do) and that the vaccination data was not linked to the data collected by the questionnaire.

18.   The sentence “Only major side effects (e.g. with an ambulance on scene) were recorded” seems out of place given that the text before this refers to satisfaction and not to side effects.

19.   The authors should clearly define what were (and how were measured) the outcomes of the study (e.g., side effects, satisfaction, vaccine administered, dose, etc).

20.   Data analysis – it is not correct to use a mean to describe an ordinal variable. As such, authors should review the analysis and compute medians and interquartile range  for satisfaction related variables Also considered presenting confidence intervals for numeric outcomes instead of the standard deviation.

21.   Please considered adding information on the number of doses administered per vaccinator as well as vaccines administered per hour/ day since this information can help planning for future MV initiatives.

22.   Consider enriching this sentence (line 221 to 223) “During the entire MVC period, only 5 ambulances were called to respond, 2 for symptoms detected prior to vaccination (bradycardia and chest pain) and 3 for post-vaccination symptoms (1 each of chest pain, hypertension, vasovagal syncope).” By computing the cumulative incidence of side effects.

23.   Organize your results in order to present:

a.       Data on the process (MCV): number of vaccines administered, number of human resources, number of hours worked, vaccines administered per vaccinator, vaccines prepared per preparator, minutes from entry point to being vaccinated, minutes between entry and exit of the MCV, minutes between stations (if available)

b.       Data on the subjects that were questioned about the satisfaction with the process

c.       Data on satisfaction.

24.   Discussion – the biggest value of the MCV in Toulouse has to do with the inclusion of health care and non-health care students in the vaccination effort and this is not discussed in the paper. During the mass vaccination that took place more or less all over the global North, the lack and/ or diversion of human resources from acute care to vaccination centers was one of the major challenges and in most situations the use of other resources such as health care students was not accepted (in some cases even the use of health care workers that usually do not vaccinate was a challenge). This paper is an opportunity to describe what enabled this solution in Toulouse and to show that despite using this additional resource, avoiding depleting other care services, is an option in epidemic emergencies. This is not fully explored in the discussion. Additionally, the authors fail to compare the performance of the MCV with other experiences.

25.   Limitations – The limitations pointed buy the authors are not limitations as they do not derive from the study/ case report in itself. The only one that should remain in this section is the one related to the response rate of the satisfaction questionnaire. The statements around the legal framework that allowed students to vaccinate and the availability of students to do so should me moved to the discussion and deepened as suggested in the previous comment.

Author Response

Point-by-point response

Reviewer 2

Dear Authors

Thank you for submitting your paper to the journal.

This paper reports the experience of setting up a MVC in Europe during the SARS-CoV-2 pandemic which I consider to be very relevant and important for future planning of this and similar interventions.

Nevertheless, the format chosen (research article) is not the most appropriate to clearly report the study. I would advise an adaptation to the case report format and I have made this suggestion to the editors. Similarly, I would advise you to consider this study as a case study instead of an observational study. A good reference for this type of study is Robert K. Yin. (2014). Case study research design and methods (5th ed.). Thousand Oaks, CA: Sage. 282 pages.

We renamed the study as a “descriptive, cross-sectional study, conducted as an intensive, systematic investigation of a new model in which we examined in-depth data relating to several variables.”

Nevertheless, the paper is well written, and it reads well. I consider that it should be considered for publication after major revision.

We thank you for your comments.

Suggested revisions:

  1. In the abstract, as well as throughout the text the author use commas to separate thousands and for decimal places. It is common use in English language to use commas to separate orders of magnitude and dots to separate decimal places. Please review accordingly.

We added comas to separate decimal places in Table 2 and Table 3.

  1. Line 56-57 . From my point of view one of the main issue that should be addressed by this type of manuscript is to specify exactly how the services were organized and designed. I would advise to specify what is meant by "Principles of disaster medicine were used in the designing of the center to ensure appropriate care despite fewer resources than normal" - What principles? Give at least one example

We gave some examples on principles of disaster medicine and discussed it in the discussion section.

  1. Line 63-64 "We performed a prospective observational study (both quantitative and qualitative) of the Toulouse MVC, from March 28th to October 20th, 2021." The authorsshould separate the type of study from the context where it was conducted. I advise to state to classify as an observation cross-sectional study since the outcomes were measured only once (althought pertaining to a period of time). Also clarify the qualitative part of the study.

We took your advice into account and added information about the chosen format.

  1. Move text in lines 66 to 69 "Data concerning the population vaccinated, the number and the type of vaccines used were recorded as part of the normal quality assurance processes of the MVC. A written satisfaction questionnaire was available in the centrefor any patient who wished to give feedback about their experience." to 2.3. Data gathering and data management.

We moved the text in lines 66 to 69 to 2.3 as suggested.

  1. The use of a), b) , c), … under “2.2. Organization…” is a little bit confusing Please check if you can use sub-headings like “2.2.1 Premises…” or avoid identations

We added sub-headings like 2.2.1; 2.2.2; 2.2.3; 2.2.4; 2.2.5

  1. Line 82 – No need for Health to have a capital H

We replaced Health by health.

  1. Please note that Figure 1 is black and white. Consider changing the type of lines used instead of using colours (e.g., dot line, doubled lines).

We modified the legend of Figure 1.

  1. Line 97 – review “in managing the human resources of the vaccinators and preparers” to “in managing the vaccinators and preparers”

We deleted human resources.

  1. Line 101 – Due a paragraph break in line 101, before “The operational coordinator oversaw the…”

We made the change.

  1. Underline or use bold to identify, the first time they are used, medical coordinator, nurse coordinator, operational coordinator, health care students and non health care students (line 89 to112).

We used bold to identify different functions.

  1. Consider presenting the roles/functions of medical coordinator, nurse coordinator, operational coordinator, health care students and non-health care students using dots for easiness of read or considered presenting a table systematizing this information

We used dots to define each function in the MVC.

  1. Line 122 to 123. The sentence “The personnel/shot ratio was defined empirically in the first months of operation and adapted in real time, and are mentioned for reference.” Breaks the reading flow. Please consider deleting or moving to a more appropriate place in the text

We moved the sentence in subheading 2.2.2 and added a table to illustrate it.

  1. Clarify where this was done Line 124-125 “Firstly, people presenting forvaccination had to show their appointment confirmation and received an informed consent letter on vaccination, including a questionnaire detailing any contraindications.” (Entrance?)

We added “at entrance” in line 232.

  1. Line 126-127 where it reads “Once signed, theywere directed to temperature control, then medical and pharmaceutical students checked their questionnaires.” It should read “Once signed, they were directed to temperature control,
    where medical and pharmaceutical students checked the contraindications questionnaire.”

We took into account your suggestion and made the change.

  1. Line 164 consider replacing “to inject vaccines” by “to administered vaccines”.

We made the change as suggested.

  1. Data gathering and data management – there are two different aspects in this paper – one that relates to the registers of the vaccination process and that the authors used later on in the text and the specific data collection of the study that pertains to a questionnaire on the quality/ satisfaction of users, not mandatory, and that the authors also use in the paper. It is important to distinguish between the two – reporting how the data was gather for the vaccination registers and how the data from the MVC was used for the purpose of this study.

We modified the presentation of the data management. Data related to the registers of the vaccination process have been moved in subheading 2.2.3, while data collection of the study are now in 2.3.

  1. The authors should state if an informed consent for the satisfaction questionnaire was obtained as well as if the study was assessed by an ethics committee. It is also important to stress that the questionnaire was anonymous (which the author already do) and that the vaccination data was not linked to the data collected by the questionnaire.

We added a paragraph about ethical consideration and informed consent in 2.3 section.

  1. The sentence “Only major side effects (e.g. with an ambulance on scene) were recorded” seems out of place given that the text before this refers to satisfaction and not to side effects.

We moved the place in of the sentence to 2.1 section.

  1. The authors should clearly define what were (and how were measured) the outcomes of the study (e.g., side effects, satisfaction, vaccine administered, dose, etc).

We described the different outcomes of the study in subheading 2.1.

  1. Data analysis – it is not correct to use a mean to describe an ordinal variable. As such, authors should review the analysis and compute medians and interquartile range  for satisfaction related variables Also considered presenting confidence intervals for numeric outcomes instead of the standard deviation.

We replaced mean by median and IQR for the overall satisfaction variable and we replaced SD by confidence interval for the age variable. All other satisfaction related variables are expressed in frequence and percentage.

  1. Please considered adding information on the number of doses administered per vaccinator as well as vaccines administered per hour/ day since this information can help planning for future MV initiatives.

We added the number of doses administered per vaccinator. The number of vaccines administered per hour was not known as opening hours changed during the duration of the study.

  1. Consider enriching this sentence (line 221 to 223) “During the entire MVC period, only 5 ambulances were called to respond, 2 for symptoms detected prior to vaccination (bradycardia and chest pain) and 3 for post-vaccination symptoms (1 each of chest pain, hypertension, vasovagal syncope).” By computing the cumulative incidence of side effects.

We added a sentence with the cumulative incidence of side effects.

  1. Organize your results in order to present: 1.Data on the process (MCV): number of vaccines administered, number of human resources, number of hours worked, vaccines administered per vaccinator, vaccines prepared per preparator, minutes from entry point to being vaccinated, minutes between entry and exit of the MCV, minutes between stations (if available) 2. Data on the subjects that were questioned about the satisfaction with the process. 3. Data on satisfaction.

We organized the results as you suggested.

  1. Discussion – the biggest value of the MCV in Toulouse has to do with the inclusion of health care and non-health care students in the vaccination effort and this is not discussed in the paper. During the mass vaccination that took place more or less all over the global North, the lack and/ or diversion of human resources from acute care to vaccination centers was one of the major challenges and in most situations the use of other resources such as health care students was not accepted (in some cases even the use of health care workers that usually do not vaccinate was a challenge). This paper is an opportunity to describe what enabled this solution in Toulouse and to show that despite using this additional resource, avoiding depleting other care services, is an option in epidemic emergencies. This is not fully explored in the discussion. Additionally, the authors fail to compare the performance of the MCV with other experiences.

We agree with your comments and enriched the discussion section.

  1. Limitations – The limitations pointed buy the authors are not limitations as they do not derive from the study/ case report in itself. The only one that should remain in this section is the one related to the response rate of the satisfaction questionnaire. The statements around the legal framework that allowed students to vaccinate and the availability of students to do so should me moved to the discussion and deepened as suggested in the previous comment.

We totally agree and deleted the statements about legal framework from the limitations section to moved in to the discussion section.